# D³RoMa: Disparity Diffusion-based Depth Sensing for Material-Agnostic Robotic Manipulation

**Songlin Wei[1,4], Haoran Geng[2,3], Jiayi Chen[1,4], Congyue Deng[3], Wenbo Cui[5,6], Chengyang Zhao[1,4], Xiaomeng Fang[6], Leonidas Guibas[3], He Wang[1,4,6]**

[1]CFCS, School of Computer Science, Peking University,
[2]University of California, Berkeley, [3]Stanford University, [4]Galbot,
[5]University of Chinese Academy of Sciences
[6]Beijing Academy of Artificial Intelligence
https://PKU-EPIC.github.io/D3RoMa

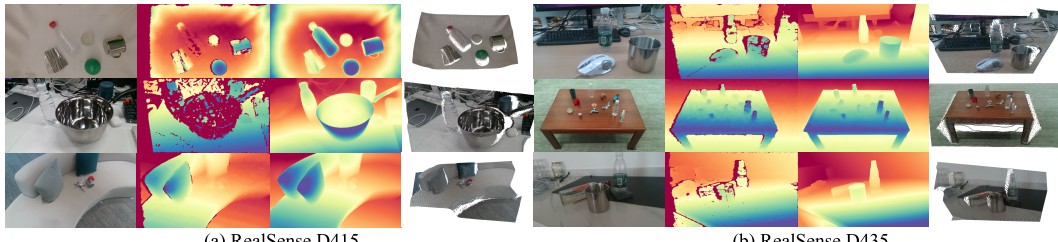

(a) RealSense D415      (b) RealSense D435

Figure 1: **Generalizability of D³RoMa in the real world.** Our method robustly predicts transparent (bottles) and specular (basin and cups) object depths in tabletop environments and beyond. RGB image, pseudo colorized raw disparity map, our prediction, and point cloud are displayed for each case of a total of 6 frames captured by camera RealSense D415 and D435. *RGB and depth images are not aligned for the D435 camera for better visualization.*

**Abstract:** Depth sensing is an important problem for 3D vision-based robotics. Yet, a real-world active stereo or ToF depth camera often produces noisy and incomplete depth which bottlenecks robot performances. In this work, we propose D³RoMa, a learning-based depth estimation framework on stereo image pairs that predicts clean and accurate depth in diverse indoor scenes, even in the most challenging scenarios with translucent or specular surfaces where classical depth sensing completely fails. Key to our method is that we unify depth estimation and restoration into an image-to-image translation problem by predicting the disparity map with a denoising diffusion probabilistic model. At inference time, we further incorporated a left-right consistency constraint as classifier guidance to the diffusion process. Our framework combines recently advanced learning-based approaches and geometric constraints from traditional stereo vision. For model training, we create a large scene-level synthetic dataset with diverse transparent and specular objects to compensate for existing tabletop datasets. The trained model can be directly applied to real-world in-the-wild scenes and achieve state-of-the-art performance in multiple public depth estimation benchmarks. Further experiments in real environments show that accurate depth prediction significantly improves robotic manipulation in various scenarios.

**Keywords:** Depth Estimation, Diffusion Model, Stereo Vision

# 1   Introduction

With the extensive use of stereo cameras, stereo depth estimation has been one of the most widely studied problems in robotics for determining the target object position or acquiring 3D information of the environment [1, 2, 3, 4]. However, the depth maps provided by existing stereo cameras suffer from severe noise, inaccuracy, and incompleteness issues, bottlenecking robot performances regardless of their well-developed recognition and manipulation algorithms.

Traditional stereo-to-depth algorithms such as SGM [5] have fundamental issues: (i) In principle, they cannot tackle non-Lambertian surfaces due to the intricate light paths; (ii) Occlusion and out-of-view areas prohibit the computation of pixel correspondences. Recent works have leveraged learning-based techniques for acquiring or restoring better depth maps [6, 7]. While they alleviate the above issues to a certain extent, predicting the depth for transparent and specular objects remains challenging as their image features from RGB pixel values are inherently ambiguous due to foreground-background color blending and thus can be misleading to correspondence estimation [8].

In this work, we propose $D^3$RoMa. Instead of building our network with cost volumes as in most prior works, we transform the dense matching problem in depth estimation into an image-to-image translation problem by predicting the disparity map with a denoising diffusion probabilistic model. Such a paradigm does not rely on low-level feature matching but rather unleashes the power of generative models to directly translate the left and right frames into the target disparity image. More concretely, our method brings twofold benefits: (i) Unlike the regression models as in prior works, the diffusion model in our framework enables generative modeling of multi-modal depth distributions for transparent or translucent surfaces. (ii) The multi-step denoising process resembles the iterative solver, replacing prior iterative networks such as RAFT-Stereo [7] and HitNet [9].

Additionally, at inference time, we further incorporate the geometric constraints from traditional stereo vision by introducing a left-right consistency loss. The loss is integrated into the diffusion sampling process as a classifier guidance. The whole paradigm combines learning-based predictions and traditional geometric modeling through a simple summation of their gradients in the score function of the diffusion model.

To train the network, we craft a synthetic dataset HSSD-IsaacSim-STD (HISS) of around $10,000$ stereo image pairs simulating real active stereo infrared patterns, including more than 350 transparent and specular objects in more than 160 indoor scenes [10]. Our dataset greatly extends the existing datasets that are limited to near-diffusive materials, table-top settings, or without realistic depth sensor simulation [11, 12, 13, 14, 15, 16]. Trained on our synthetic dataset, our model can be directly applied to real-world in-the-wild scenes (Figure 1) and achieve state-of-the-art performances not only on traditional stereo benchmarks but also on datasets targeting specular, transparent, and diffusive (STD) objects. To further validate our effectiveness in robotic manipulation, we conduct experiments on both simulated and real environments ranging from tabletop grasping to mobile grasping in indoor scenes. We observe that with the high-quality depth maps and 3D point clouds predicted by our method, the success rates of robotic manipulation can be significantly improved in diverse settings.

To summarize, our contributions are: (1) A diffusion model-based stereo depth estimation framework that can predict state-of-the-art depth and restore noisy depth maps for transparent and specular surfaces; (2) An integration of stereo geometry constraints into the learning paradigm via guided diffusion; (3) A new scene-level STD synthetic dataset that simulates real depth sensor IR patterns and photo-realistic renderings; (4) Significant improvements in robotic manipulation tasks with our higher-quality depth maps and 3D point clouds.

# 2   Related Work

**Stereo Depth Estimation and Completion.**   Modern deep-learning stereo methods [7, 17] typically have the following structures. First, a feature encoder are used to extract the left and right

image features. The feature encoders are either pre-trained and frozen or trained end-to-end. Second, a cost volume is built by enumerating all the possible matching. Some works incorporate 3D CNN or attention networks to increase the receptive field of convolution layers which have proved to be beneficial [18, 19]. Finally, a detection head is added to regress the disparities. The most successful ingredient is the iterative mechanism which is proposed by the seminar work RAFT [6] [7]. On the other hand, Weinzaepfel et al. [20] uses cross-view completion pertaining and achieves impressive results. Despite all the progress that has been made, in the real world, transparent and specular objects are ubiquitous and the RGB features of the surfaces are inherently ambiguous to be used to search for correspondence because the foreground and background colors are blended.

Previous work [13] tried to restore the missing depths with the help of neighboring raw depth values and their RGB color clues [12] [21] but with limited generalizability. Another line of work directly fine-tunes a trained deep stereo network [14] on transparent surfaces allowing the feature encoders to learn to match the transparent surfaces. However, the aforementioned foreground and background ambiguities confuse the feature-matching-based pipeline when dealing with regular diffuse objects.

**Diffusion Model for Depth Estimation.**   Recently, researchers have employed diffusion models to estimate optical flow [22] and predict depths with a single RGB image input [23] [24] [25] [26]. Such monocular methods can estimate the depth map up to an unknown absolute scale. Bhat et al. [27] proposed to train an extraneous network to predict the scale and achieve decent accuracy. However, the monocular methods either lack the absolute scale or have inferior accuracy for robotic manipulation tasks. Another line of work has pioneered adjusting the diffusion model to stereo settings. Nam et al. [28] proposed to learn matching based on cost volume in a diffusion manner, which could not handle matching ambiguities. Shao et al. [29] proposed to refine raw map for high quality human reconstruction. The authors designed a novel linear scheduler and condition on all the stereo-related information. Nonetheless, we found that using the default DDPM [30] scheduling works well and only needs to condition on stereo images and raw disparity if necessary. Additionally, we combined the stereo matching loss gradient with the learned gradient by the diffusion model. The stereo matching loss is obtained by checking the left-right image photometric consistency in an unsupervised manner [31] [32]. Such guided diffusion model [33] achieves the best results in our experiments.

**3D Vision-Based Robotic Manipulation.**   3D vision is becoming increasingly critical for robotic manipulation [34]. Most basically, depth perception enables robots to comprehend the size, shape, and position of objects within three-dimensional space, thereby facilitating more sophisticated and reliable interactions [35, 36]. Moreover, numerous works [37, 38, 39, 40] utilize RGB-D point clouds as input. However, the substantial domain gap between simulated and real RGB-D images can result in a significant sim-to-real gap [41]. Additionally, transparent and specular objects exacerbate this issue, leading to poor depth-sensing performance. Policies trained in simulators often struggle to transfer effectively to real-world scenarios, particularly in the context of mobile manipulation. Our proposed high-performance depth estimation network is a promising direction to improve existing 3D vision-based tasks.

## 3   Method

In this section, we introduce $\mathbf{D}^3\mathbf{RoMa}$, a disparity diffusion-based depth sensing framework for material-agnostic robotic manipulation. Our framework focuses on improving the accuracy of disparity map in depth estimation, especially for transparent and specular objects which are ubiquitous yet challenging in robotic manipulation tasks. Given an observation of the scene, our framework takes the raw disparity map $\tilde{D}$ and the left-right stereo image pair $I_l, I_r$ from the depth sensor as input, and outputs a restored disparity map $x_0$, which will be converted into the restored depth map.

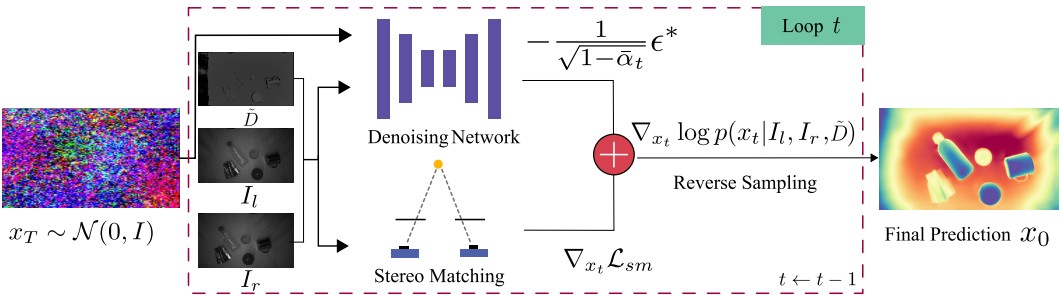

Figure 2: **Disparity diffusion with stereo-geometry guidance.** Our disparity diffusion-based depth sensing framework takes the raw disparity map $\tilde{D}$ and the left-right stereo image pair $I_l, I_r$ as input. With the geometry prior from the stereo matching between $I_l$ and $I_r$ as guidance for the reverse sampling, our diffusion model can gradually perform the denoising process conditioned on $\tilde{D}$ to predict the restored disparity map $x_0$.

## 3.1 Preliminaries

**Stereo Vision and Depth Estimation.** Once the disparity map $x$ for the observed points between a pair of stereo cameras is known, the depth map $d$ for the points can then be calculated using the camera intrinsic parameters through $d = (f \cdot b)/x$, where $f$ and $b$ are the camera focal length and the stereo baseline, respectively. The estimation of the disparity map $x$ is traditionally modeled as a dense matching problem, which can be solved within the image domain. Thus, stereo depth estimation can be studied independently from different camera devices.

**Denoising Diffusion Probabilistic Model.** Diffusion models [30] [42] are special latent variable models that reverse the diffusion (forward) process which gradually diffuses the original data $x_0$ through a Markov process. When the network $s_\theta$ is trained converged, the gradient of the noise distribution also called the score function [43] is

$$\nabla x_t \log p(x_t) \approx -\frac{1}{\sqrt{1 - \bar{\alpha}_t}} s_{\theta^*}(x_t, t; \theta). \tag{1}$$

During inference, data samples can be generated through ancestral sampling which resembles the stochastic gradient Langevin dynamics (SGLD) [44]

$$x_{t-1} = \frac{1}{\sqrt{1 - \beta_t}} (x_t + \beta_t \nabla_{x_t} \log p(x_t)) + \beta_t \epsilon_t, \epsilon_t \sim \mathcal{N}(0, I). \tag{2}$$

## 3.2 Disparity Diffusion for Depth Estimation

In this work, we formulate the stereo depth estimation problem as an image-to-image translation problem in the diffusion model. One important design choice is what to condition on. The model is usually formulated to condition on the stereo image pairs $I_l, I_r$ for stereo depth estimation. Our experiments found that conditioning additionally on raw disparity $\tilde{D}$ makes the network converge faster during training and generalizes more robustly in out-of-distribution scenarios. The raw disparity can be easily obtained either from a traditional stereo matching algorithm SGM [5] or from the real camera sensor outputs. For real active stereo depth sensors like RealSense, the left and right images are captured by infrared (IR) cameras with special shadow patterns projected by an IR projector. As a result, conditioning on the left and right images and the raw disparity map $\tilde{D}$, we train a conditional diffusion model to learn the distribution of the disparity map

$$p_\theta(x_0|y) = \int p_\theta(x_{0:T}|y)dx_{1:T}, \quad p(x_{0:T}|y) = p(x_T) \prod_{t=1}^{T} p_\theta(x_{t-1}|x_t, y) \tag{3}$$

where $y = \{I_l, I_r, \tilde{D}\}$. Empirically, this conditional denoising network has been shown to be successful [45] [46]. Batzolis et al. [47] further provided proof (see Theorem 1) that the conditional

score $\nabla_{x_t} p(x_t|y)$ can be learned through the same training objectives as in the unconditional case even though the condition $y$ does not appear in the training objectives. After the network is trained, the estimated disparity can then be sampled through

$$x_{t-1}|y = \frac{1}{\sqrt{1-\beta_t}}(x_t + \beta_t \nabla_{x_t} \log p(x_t|y)) + \beta_t \epsilon_t, \epsilon_t \sim \mathcal{N}(0, I). \tag{4}$$

### 3.3 Reverse Sampling Guided by Stereo Geometry

Inspired by the classier guidance to image generation tasks [48] [49], we propose to guide the disparity diffusion process with the model-based geometry gradient. The guided reverse process is illustrated in Figure 2. Specifically, the conditional score function is perturbed with gradient computed from stereo-matching

$$\nabla_{x_t} \log p(x_t|y) = -\frac{1}{\sqrt{1-\bar{\alpha}_t}} s_{\theta*}(x_t, t, y; \theta) + s\nabla_{x_t} \mathcal{L}_{\text{sm}}(I_l, I_r, x_t) \tag{5}$$

where $\mathcal{L}_{\text{sm}}$ is the similarity loss function which compares the left image with the warped left image. The warped left image is obtained by warping the right image with the estimated disparity. $s \in \mathbb{R}^+$ controls the geometric guidance strength and it balances the learned gradient from diffusion model and geometric gradient from stereo models. A detailed derivation of Equation 5 is provided in Appendix B. To mitigate the gradient locality in stereo matching, we downsample the stereo images into multiple different lower resolutions when computing the gradient of stereo matching. More specifically, we have

$$\mathcal{L}_{\text{sm}}(I_l, I_r, x_t) = \sum_k \mathcal{L}_{\text{ssim}}(I_l, I_r, x_t) + \gamma \mathcal{L}_{\text{smooth}}(I_l, x_t) \tag{6}$$

where $k$ is the index of the layer for different resolutions and $\gamma \in \mathbb{R}^+$ is a weighting constant balances the two photometric and smoothness losses. The $\mathcal{L}_{\text{ssim}}$ is the structural similarity index (SSIM) [50] which computes the photometric loss between the left image $I_l$ and warped image $\tilde{I}_{\text{left}}$:

$$\mathcal{L}_{\text{ssim}}(I_l, I_r, x_t) = \text{SSIM}(I_l, \tilde{I}_l), \tag{7}$$

$$\tilde{I}_l(u, v) = I_r\langle u + x_t, v\rangle \tag{8}$$

where $u, v$ are the pixel coordinates in the image plane and $\langle \rangle$ is linear sampling operation. $\mathcal{L}_{\text{smooth}}$ is an edge-aware smoothness loss [31] [32] defined as

$$\mathcal{L}_{\text{smooth}}(I_l, x_t) = |\partial_u x_t|e^{-|\partial_u I_l|} \tag{9}$$

which regularizes the disparity by penalizing large discontinuity in non-edge areas. Here $\partial_u$ means partial derivative in $u$ (horizontal) direction in the image plane. Then we predict the disparity map $x_0$ with the perturbed gradient from Equation 5 following the sampling process introduced in Equation 2. Finally, We can convert the disparity to depth once we know the camera parameters.

### 3.4 HISS Synthetic Dataset

We create our synthetic dataset **HISS** based on Habitat Synthetic Scenes Dataset (HSSD) [10]. We leverage the 168 high-quality indoor scenes from HSSD to increase scene diversity. For objects, we include a total of over 350 object models from DREDS [13] and GraspNet [34]. The scene and randomly selected object CAD models are rendered in Isaac Sim [51]. During rendering, object materials and scene lighting are specifically randomized in simulation to mimic the transparent or specular physical properties of objects (cups, glasses, bottles, *etc.*) in the real world. To obtain the correct depth values for transparent surfaces, we adopt a two-pass methodology. We first render RGB images and depth maps of the scenes with object materials set to diffuse. The lightings are all turned to enable photorealistic rendering. In the second pass, we turn the normal lighting off and project a similar shadow pattern on the scenes to mimic the RealSense D415 infrared stereo images. Using the intrinsics of the RealSense D415 depth camera, we render over 10,000 photo-realistic stereo images with simulated shadow patterns. Experiments demonstrate that our dataset is the key enabler of our method's excellent generalizability in the real world.

# 4 Experiments

## 4.1 Depth Estimation in Robotic Scenarios

**DREDS.** We evaluate our method on DREDS [13], a tabletop-level depth dataset with both synthetic and annotated real data for specular and transparent objects. We compare our method with several state-of-the-art baselines: 1) NLSPN, 2) LIDF, 3) SwinDR, and 4) ASGrasp, which are four methods that have been shown effective on depth estimation for transparent or specular objects. We also compare another 3 stereo depth SoTA methods: RAFT-Stereo, IGEV-Stereo, and CroCo-Stereo. We use mean absolute error (MAE), relative depth error (REL), root mean square error (RMSE), along with 3 other metrics related to depth accuracy as metrics for evaluation. We include detailed descriptions for the baselines and the metrics in the Appendix C.

As shown in Table 1, on the DREDS-CatKnown data split (synthetic data), all variants of our method surpass all baselines on all metrics. Further, the results of our ablations show that the performance of our method can be steadily improved with more information provided, especially the integration of the raw disparity.

Table 1: **Comparisons of Depth Estimation Results on DREDS Dataset (DREDS-CatKnown split, synthetic).** We also studied different combinations of conditioned images for the denosing network. Best shown in **bold** and second best shown in underlined.

| Methods | RMSE $\downarrow$ | REL $\downarrow$ | MAE $\downarrow$ | $\delta_{1.05} \uparrow$ | $\delta_{1.10} \uparrow$ | $\delta_{1.25} \uparrow$ |
|---|---|---|---|---|---|---|
| NLSPN (Park et al. [21]) | 0.010 | 0.009 | 0.006 | 97.48 | 99.51 | 99.97 |
| LIDF (Zhu et al. [12]) | 0.016 | 0.018 | 0.011 | 93.60 | 98.71 | 99.92 |
| SwinDR (Dai et al. [13]) | 0.010 | 0.008 | 0.005 | 98.04 | 99.62 | 99.98 |
| ASGrasp (Shi et al. [14]) | 0.007 | 0.006 | 0.004 | - | - | - |
| RAFT-Stereo (Lipson et al. [7]) | 0.007 | 0.006 | 0.005 | 98.13 | 99.83 | 99.97 |
| IGEV-Stereo (Xu et al. [18]) | 0.006 | 0.007 | 0.002 | 98.19 | 99.66 | 99.97 |
| CroCo-Stereo (Weinzaepfel et al. [52]) | 0.008 | 0.010 | 0.002 | 94.49 | 98.32 | 99.87 |
| D$^3$RoMa(Cond. on RGB+Raw) | 0.0045 | 0.0016 | 0.0011 | 99.64 | 99.88 | **99.99** |
| D$^3$RoMa(Cond. on RGB+Left+Right) | 0.0070 | 0.0048 | 0.0032 | 99.11 | 99.79 | 99.98 |
| D$^3$RoMa(Cond. on RGB+Left+Right+Raw) | **0.0040** | **0.0014** | **0.0010** | **99.71** | **99.90** | **99.99** |

More ablation studies regarding the geometry-based guidance, network hyper-parameters and architectures are provided in Appendix F.

**SynTODD.** SynTODD [15] is another synthetic dataset for transparent objects by using Blender. It contains 87512 train images and 5263 test images. The authors proposed a novel multi-view method (MVTrans) to estimate object depths, poses, and segmentations. Because the dataset provides neither simulated raw disparity nor correct camera intrinsic for stereo images, We compare with MVTrans[15] using our monocular variant, ie., we modify the network to condition only on RGB images. We do scale alignment after the prediction of the scale-invariant depth following MiDas [53]. As shown in Table 2, Our method achieves better performance than all the variants of MVTrans including 2-views, 3-views, and 5-views.

**ClearPose** ClearPose [16] is a large-scale real-world RGB-D benchmark for transparent and translucent objects. The dataset contains 350,000 real images captured by the RealSense L515 depth camera. The authors collected a set of very challenging scenes including different backgrounds, heavy occlusions, objects in translucent and opaque covers, on non-planar surfaces, and even filled with liquid. We evaluate our method on ClearPose in all settings. Our method D$^3$RoMa outperforms two previous SoTA ImplicitDepth [54] and TransCG [55] by a large margin as shown in Table 3. More detailed statistics of each scene are provided in Table 11 and Figure 10 in Appendix H. ClearPose is captured with RealSense L515 camera, the depth noise is large when point distance is larger than 5 meters. We mask out the noise depth values out of the range [0.2,5] meters for all the experiments.

Table 2: **Depth estimation results on Syn-TODD.**

| Methods | RMSE ↓ | REL ↓ | MAE ↓ |
|---|---|---|---|
| SimNet(Laskey et al. [56]) | 1.229 | 0.975 | 1.020 |
| MVTrans (2 images) | 0.134 | 0.135 | 0.089 |
| MVTrans (3 images) | 0.125 | 0.125 | 0.083 |
| MVTrans (5 images) | 0.124 | 0.117 | 0.080 |
| D$^3$RoMa(Cond. on RGB) | **0.065** | **0.079** | **0.040** |

Table 3: **Results on Depth completion benchmark ClearPose.**

| Methods | RMSE↓ | REL ↓ | MAE ↓ |
|---|---|---|---|
| ImplicitDepth | 0.133 | 0.120 | 0.101 |
| TransCG | **0.077** | 0.065 | 0.060 |
| D3RoMa (Cond. On RGB+Raw) | 0.093 | **0.031** | **0.412** |

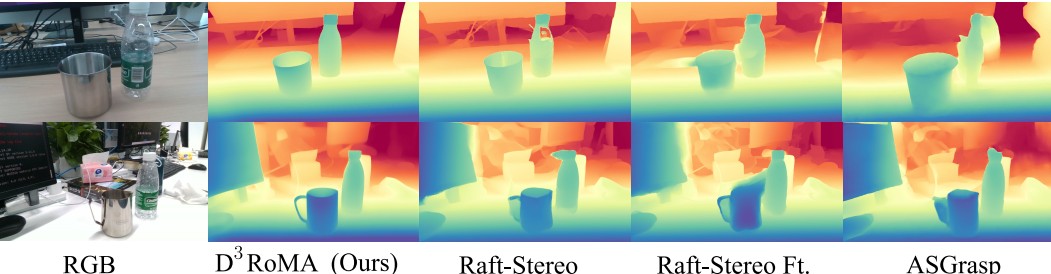

RGB     D$^3$RoMA (Ours)     Raft-Stereo     Raft-Stereo Ft.     ASGrasp

Figure 3: **Qualitative Comparisons with SOTA methods in the Real World**. Each row (from left to right) shows the RGB image and disparity results of our method, pre-trained Raft Stereo, Raft Stereo fine-tuned on our dataset, and ASGrasp.

## 4.2 Comparisons with SOTA Stereo Matching Methods in General Scenarios

We further demonstrate the effectiveness of our method for stereo matching in general scenarios. We compare our method with state-of-the-art stereo-matching baselines on SceneFlow [57], a synthetic dataset containing more than 39,000 stereo frames in $960 \times 540$ pixel resolution. The dataset contains three challenging scenes, FlyingThings3D, Driving, and Monkaa, which makes it a high-quality dataset for pre-training [7, 18, 20]. We train our model from scratch using 35,454 stereo pairs for training and leave the rest as testing split. We also resize the images in the dataset into $480 \times 270$ to be consistent with our robotic perception settings. Following the previous work [18], the ground truth disparity is normalized using the maximum disparity value 192, which is also used to crop the test data. More implementation details are provided in Appendix E. As shown in Table 4, we achieve the best results compared to existing state-of-the-art methods.

Table 4: **Comparisons with SOTA Stereo Matching Methods on SceneFlow Dataset.** $^*$Ours is not built on Cost Volumes.

| Methods | GA-Net [58] | LEAStereo [59] | EdgeStereo [60] |
|---|---|---|---|
| EPE | 0.84 | 0.78 | 0.74 |
| ACVNet[61] | IGEV-Stereo[18] | HitNet [9] | D$^3$RoMa$^*$ |
| 0.48 | 0.47 | **0.36** | **0.36** |

## 4.3 Robotic Manipulation

In the grasping experiments, we first acquire the depth map by $D = (f \cdot b)/X$. Then back project the depth into point cloud $\mathcal{P} = DK^{-1}P$, where $K \in \mathbb{R}^{3 \times 3}$ is the camera intrinsics and $P$ are the homogeneous points in the image plane corresponding to each pixel. With the restored point cloud, we leverage GSNet [62] to predict 6 DoF grasping poses. To increase the grasping success rate for all baselines, we filter the grasping pose which has the angle between the grasping pose and the $z$ (up) direction less than 30 degrees. We always select the grasping pose with the highest core and transform it into the robot base frame.

**Environment Setup.** We set up a tabletop grasping, articulated object manipulation and a mobile grasping environment in the real world, as shown in Figure 4. In the tabletop grasping experiments,

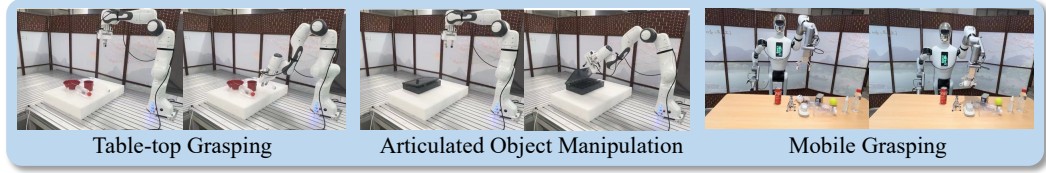

| Table-top Grasping | Articulated Object Manipulation | Mobile Grasping |

Figure 4: **Robotic Manipulation.** We examine our approach on challenging robot manipulation tasks and our performance significantly outperforms all baselines.

we use a Franka 7-DoF robotic arm. We place STD objects on surfaces with bumps and pits, which are challenging setups for both depth sensing and robot grasping. The objects have non-diffusive surface materials such as glass, porcelain, glass, etc.

In the mobile grasping experiments, we use a customized wheeled robot with 7 DoF arms as shown in Appendix E.3. The robot is arbitrarily placed near the target objects omitting the navigation phase which is beyond our scope. We place the objects in clusters at different tables and furniture with both flat and non-flat surfaces and different backgrounds, bringing challenges to depth sensing with the complex scene environments. More details about the articulated object manipulation experiments can be found in Appendix J.

**Results and Analysis.** We compare our method with two other baselines. All baselines use the same motion planner CuRobo [63] but different depth sensing. We also compare with ASGrasp [14] which was mainly designed for table-top grasping of STD objects. We report the results for different objects (STD) respectively and a overall success rate in Table 6. The quantitative results for three different scenes of mobile manipulation are provided in Table 5. While ASGrasp and $D^3$RoMa both improved over the raw sensor outputs, our method outperforms ASGrasp with a large margin.

Table 5: Mobile grasping success rate of different baselines with the same motion planner in real environments. Each cell shows SR on specular, transparent, and diffusive objects separated with /.

| Baselines | Tea Table | Kitchen Table | Sofa |
|---|---|---|---|
| Raw | 0.40/0.22/0.67 | 0.67/0.63/1.00 | 0.50/0.75/0.67 |
| ASGrasp | 0.60/0.89/0.83 | 0.67/0.75/0.83 | 0.67/0.875/0.83 |
| $D^3$RoMa | **0.80/1.00/1.00** | **0.67/0.88/1.00** | **0.83 / 0.875 / 0.83** |

Table 6: Comparisons of tabletop grasping Success Rate (SR) with different depth sources. S. = specular, T. = transparent, D. = diffusive.

| Baselines | S. | T. | D. | Overall |
|---|---|---|---|---|
| Raw | 0.33 | 0.25 | 0.70 | 0.45 |
| ASGrasp | 0.63 | 0.43 | 0.50 | 0.50 |
| $D^3$RoMa | **0.83** | **0.63** | **0.78** | **0.77** |

## 5 Conclusion

In this work, we proposed a novel geometry gradient guidance to diffusion model in disparity space to predict depth for stereo images. Conditioning on stereo image pair and a raw disparity map, our network achieves SOTA performance on existing benchmarks. Both disparity evaluation on pure synthetic datasets and depth evaluation on depth datasets demonstrate the efficiency of our method. Our key observations include data diversity can make a big impact on generalization in the wild and guidance helps in more challenging real-world scenes. Current 3D vision-based robotics manipulation pipelines including grasping and part manipulation can be significantly improved simply by improved depth perception. Especially, we found that depth estimation for challenging transparent objects could be better dealt with generative models than traditional stereo methods.

**Limitations and Future Work** One limitation inherited from the diffusion model is the iterative inference with many denoising steps. This issue can be mitigated by incorporating diffusion sampling acceleration techniques studied in a variety of existing works. Another limitation is that the EPE is linear to the resolution of the input image. Future research could use coarse-to-fine or simple sliced techniques to split high resolution images into smaller patches to resolve this issue.

**Acknowledgments**

This work is supported by the National Natural Science Foundation of China(No.62306016).

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

## Supplementary Materials

## A   Overview

In this supplementary material, we will give the derivation for geometry guided diffusion model for stereo vision in Appendix B, introduce different metrics in Appendix C, study some interesting properties of our method in Appendix D, provide more implementation details in Appendix E, analyze different guidance modes in Appendix F, study alternative guidance in Appendix G, show more results of our methods in Appendix H, provide more data samples in our dataset **HISS** in Appendix I, and perform more experiments of mobile part manipulation in Appendix J.

## B   Geometry Guidance for Stereo Vision

To complement the main body of the paper, we provide the detailed derivation of the geometry guided diffusion model which appears in Equation 9 in the main text.

### B.1   Stereo Vision

We define $y = \{I_l, I_r\}$ represents the conditioning stereo image pair and $x_t$ is the noisy depth at time step $t$. By Bayes' theorem, we have

$$p(x_t|y) = \frac{p(x_t)p(y|x_t)}{p(y)} \tag{10}$$

$$\log p(x_t|y) = \log p(x_t) + \log p(y|x_t) - \log p(y) \tag{11}$$

Task derivative with respect to $x_t$ on both sides of Equation 11:

$$\nabla x_t \log p(x_t|y) = \nabla x_t \log p(x_t) + \nabla x_t \log p(y|x_t) \tag{12}$$

Now, partition the second term $\log p(y|x_t)$ as

$$\begin{aligned} \log p(y|x_t) &= \log p(I_l, I_r|x_t) \\ &= \log p(I_l|x_t) + \log p(I_r|I_l, x_t) \\ &= \log p(x_t|I_l) + \log p(I_l) - \log p(x_t) + \log p(I_r|I_l, x_t) \end{aligned} \tag{13}$$

where we apply Bayes' theorem again in the third equation. Substitute Equation 13 back to Equation 12, we have

$$\nabla x_t \log p(x_t|y) = \nabla x_t \log p(x_t|I_l) + \nabla x_t \log p(I_r|I_l, x_t) \tag{14}$$

The first term is learned by the denoising network and the second term is the geometric guidance which can be calculated by stereo matching. In the experiments, we leverage more available data such as $I_r$ and $\tilde{D}$ in addition to $I_l$ into the network during training:

$$\nabla_{x_t} \log p(x_t|y) = -\frac{1}{\sqrt{1-\bar{\alpha}_t}} s_{\theta^*}(x_t, t, y; \theta) + s\nabla_{x_t}\mathcal{L}_{\text{sm}}(I_l, I_r, x_t) \tag{15}$$

Here we empirically scale the geometry gradient with $s \in \mathbb{R}^+$ and set it to 1 in the experiments.

## B.2 Extend to Active Stereo Vision

In addition to the left and right IR images, active stereo cameras provide another color image $I_c$ captured from a third color camera. While the above derivation directly applies to active stereo cameras if we ignore the color image, we found that further feeding the color image into the network slightly improves the performance in DREDS [13]. However, most stereo datasets are *passive* and do not have additional color images. Therefore, during mixed dataset training, this additional color image is dropped. Here, we provide an active stereo version of derivation analogous to Equation 13:

$$
\begin{aligned}
\log p(y|x_t) &= \log p(I_c, I_l, I_r|x_t) \\
&= \log p(I_c|x_t) + \log p(I_l|I_c, x_t) + \log p(I_r|I_l, I_c, x_t) \\
&= \log p(I_c|x_t) + \log p(I_r|I_l, x_t) \\
&= \log p(x_t|I_c) + \log p(I_c) - \log p(x_t) + \log p(I_r|I_l, x_t)
\end{aligned}
\tag{16}
$$

where the third equation assumes $p(I_l|I_c, x_t) = 1$. The $I_c$ and $I_l$ are already aligned and the only difference is the shadow pattern projected from the camera IR projector. The shadow pattern is irrelevant to the depth. Therefore, $I_c$ is approximately the sufficient statistic of $I_l$. For the same argument, we have $\log p(I_r|I_l, x_t) = p(I_r|I_l, I_c, x_t)$. Likewise, the guidance for the active stereo camera can then be obtained by substituting Equation 16 into Equation 12:

$$\nabla_{x_t} \log p(x_t|y) = \nabla_{x_t} \log p(x_t|I_c) + \nabla_{x_t} \log p(I_r|I_l, x_t) \tag{17}$$

In active stereo vision scenarios, we further train the network by conditioning it also on other available images. We set $y = \{I_l, I_r, I_c, \tilde{D}\}$.

## C  Baselines and Metrics

**Baselines.**  **NLSPN** [21] is a depth completion work that uses an end-to-end non-local spatial propagation network to predict dense depth given sparse inputs. **LIDF** [12] proposes to learn an implicit density field that can recover missing depth given noisy RGB-D input. **SwinDR** [13] proposes a depth restoration framework based on SWIN transformer and is trained on a proposed table-top dataset with STD objects (DREDS). **ASGrasp** [14] proposes a stereo-depth estimation method based on Raft-Stereo to predict two-layer depths for tabletop grasping. **Raft-Stereo** [7] is the seminal deep stereo network. To this day, it is still the most adopted architecture in stereo vision.

**Disparity Metric.**  End-Point Error (**EPE**) $= \frac{1}{H \times W} \sum |X - \hat{X}|$ is the mean absolute difference for all pixels between the ground truth and estimated disparity map.

**Depth Metrics.**  We use the following depth metrics: 1) **RMSE** $= \sqrt{\frac{1}{H \times W}|D - \hat{D}|^2}$ is the root mean square error between ground truth and predicted depths, 2) **MAE** $= \frac{1}{H \times W}|D - \hat{D}|$ is the mean absolute depth error, 3) **REL** $= \frac{1}{H \times W}|D - \hat{D}|/D$ is the mean absolute relative difference, and 4) accuracy metric $\boldsymbol{\delta_i}$ is the percentage of pixels satisfying $\max(\frac{d}{\hat{d}}, \frac{\hat{d}}{d}) < \delta_i$ where $\delta_i \in \{1.05, 1.10, 1.25\}$.

# D Interesting Properties of Generative Stereo Vision

## D.1 Uncertainty Estimation

Because our method is diffusion model based, we inherited the stochasticity in the reverse sampling process. To visualize the stochasticity, we run the same input 10 times. The uncertainty is obtained as the variance of the output disparity map. We conduct the experiments on DREDS and show the results in Figure 5. We observed that high uncertainty area corresponds to object edges where depth dramatically changes between foreground and background. Flat surfaces have lower uncertainty as the geometry is simpler. Such uncertainty could be used to filter outliers.

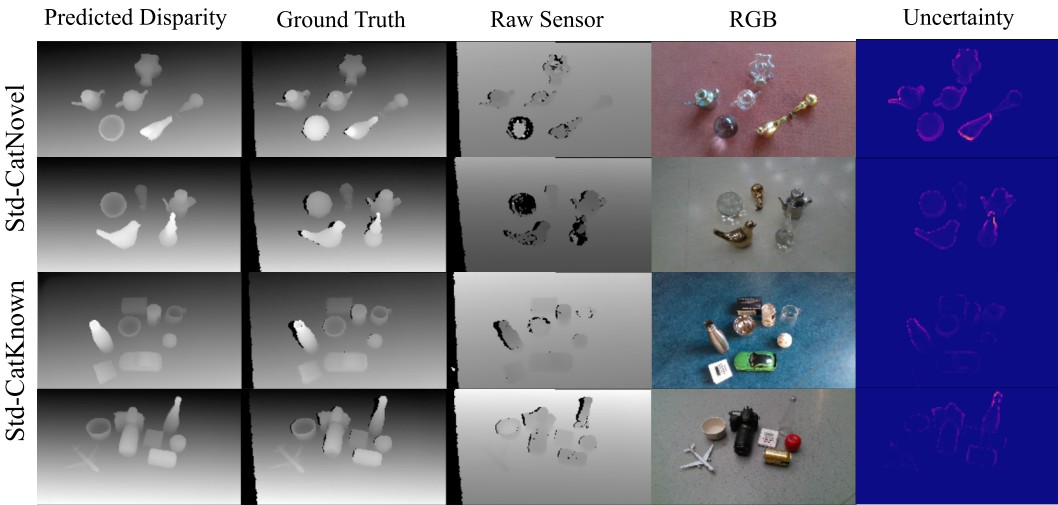

Figure 5: We visualize sample variance as uncertainty in the last column.

## D.2 Generalization Comparisons with Monocular Methods

While our method works only in stereo cases, there are seminar works predicting depth given single RBG images. The attractive part of monocular depth estimation (MDE) is that more data is available for training. Therefore, these methods can be generalized well in the wild. While some monocular methods like ZeoDepth [27] propose to recover metric depth after a special training procedure, most monocular methods predict relative depth. The relative depth can be recovered with an absolute scale which can be obtained via other sensors like lidar or prior knowledge. However, our experiments (Figure 6) found that most monocular methods produce inferior quality depth even without considering the absolute scale.

# E Implementation Details

## E.1 CUDA accelerated Semi-Global Matching

We used libSGM[64], a CUDA-accelerated, widely adopted implementation of the Semi-Global Matching (SGM)[5] algorithm. To seamlessly integrate libSGM into our pipeline, we utilized pybind11 to encapsulate the original codebase within our Python-based framework. This integration allows the adapted version of libSGM to achieve a performance of approximately 55 frames per second (FPS) at an input resolution of $960 \times 540$, with around 380MB of memory allocated on an NVIDIA RTX 4090 GPU.

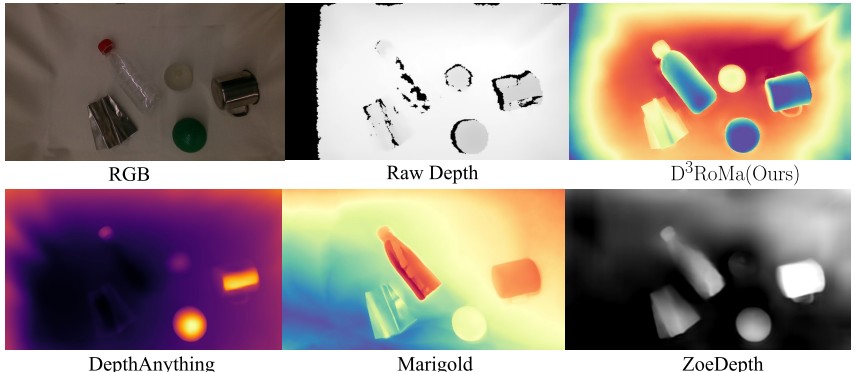

| RGB | Raw Depth | D$^3$RoMa(Ours) |
| --- | --- | --- |

| DepthAnything | Marigold | ZoeDepth |
| --- | --- | --- |

Figure 6: Generalization comparisons with State-of-the-art *monocular* depth estimation methods. All the results except ours are taken from their official web demo. Different methods used different color maps.

## E.2 Network HyperParameters and Training

We implement our network using Hugging Face Diffusers [65] and pre-compute raw disparity maps using libSGM [64]. The network is trained 600 epochs with the batch size 6×8 and a constant learning rate 0.0001. All the images are randomly cropped to 320×240 and no other data augmentation is used during training. We use cosine scheduler [66] with 128 denoising time steps for $\beta_t$ starting at 0.0001 and ending at 0.02. We use UNet as our denoising network. In the DREDS experiments, we have 6 downsampling ResNet blocks each layer has 128, 128, 256, 256, 512, and 512 channels. The second-to-last channel is a downsampling block with spatial attention. We use MSE as our loss function. For the SceneFlow experiment, we scale down the original image resolutions from 960×640 into 480×270. We use a multi-resolution pyramid noise strategy as in [26]. We further use pretrained StableDiffusion v2 [67] in the grasping experiments and adapt the input Conv block accordingly to the conditioning inputs [26]. We also train the mixed datasets including DREDS, HISS, and SceneFlow at the batch level.

## E.3 Grasping Implementation and Hardware Setup

In the grasping experiments, we mount the RealSense D415 on the wrist of the arm. After the camera captures a frame, we first acquire the depth map by $D = (f \cdot b)/X$. Then back project the depth into point cloud $\mathcal{P} = DK^{-1}P$, where $K \in \mathbb{R}^{3\times3}$ is the camera intrinsics and $P$ are the homogeneous points in the image plane corresponding to each pixel. With the restored point cloud, we leverage GSNet [62] to predict 6 DoF grasping poses. To increase the grasping success rate for all baselines, we filter the grasping pose which has the angle between the grasping pose and the $z$ (up) direction less than 30 degrees. We always select the grasping pose with the highest core and transform it into the robot base frame. Then we grasp the object with a motion planner like CuRobo [63]. *We did not perform workspace point cloud cropping operation as in the baseline ASGrasp [14] hence leading to an overall success rate drop in the main text compared with the numbers reported in ASGrasp.*

We use a wheeled mobile base mounted with two 7 DoF customized arms in the real mobile grasping experiments. Each arm attaches a parallel gripper. We only use the left arm in the experiments. Figure 7 displays the robot and the workplace.

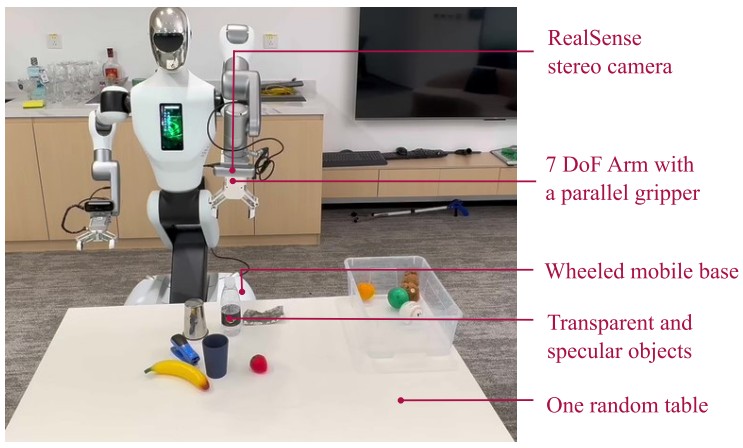

Figure 7: The robot used in the real mobile grasping experiments.

# F    Ablations Studies

## F.1    Ablations on Geometry-Guidance

Since DREDS does not provide IR images for the STD-CatKnown and STD-CatNovel data split (real data), we train the variant of our framework which only conditions on RGB image and raw disparity to compare with SwinDR.

We further perform ablation studies for the geometry-based guidance on the STD-CatKnown and STD-CatNovel data split to validate its effectiveness for diffusion-based depth estimation in real-world scenarios. As Table 7 shows, our geometry-based guidance can significantly boost performance, especially for out-of-distribution scenarios. More ablation studies regarding network hyperparameters and architectures are provided in Appendix F.

Table 7: **Evaluations of Geometry Guidance on DREDS Dataset (STD-CatKnown and STD-CatNovel split, real).** Ground truth depth is cropped in range $[0.2, 2]$.

| Methods | Guidance | RMSE ↓ | REL ↓ | MAE ↓ | $\delta_{1.05}$ ↑ | $\delta_{1.10}$ ↑ | $\delta_{1.25}$ ↑ |
|---|---|---|---|---|---|---|---|
| | | STD-CatKnown | | | | | |
| SwinDR (Dai et al. [13]) | | 0.015 | 0.013 | 0.008 | 96.66 | 99.03 | 99.92 |
| D³RoMa(Cond. on RGB+Raw) | × | 0.0109 | 0.0051 | 0.0036 | 98.41 | 99.46 | **99.94** |
| D³RoMa(Cond. on RGB+Raw) | ✓ | **0.0101** | **0.0042** | **0.0030** | **99.03** | **99.57** | 99.93 |
| | | STD-CatNovel | | | | | |
| SwinDR (Dai et al. [13]) | | **0.025** | 0.033 | 0.017 | 81.55 | 93.10 | **99.84** |
| D³RoMa(Cond. on RGB+Raw) | × | 0.0390 | 0.0177 | 0.0104 | 91.19 | 96.17 | 99.51 |
| D³RoMa(Cond. on RGB+Raw) | ✓ | 0.0397 | **0.0158** | **0.0092** | **92.78** | **97.13** | 99.61 |

## F.2    Ablations on Network HyperParameters and Architectures

We provide ablation studies on the DREDS dataset in Table 8. The baseline is conditioned on the left, and right image and raw disparity. Its hyperparameters and network architecture are described in Appendix E.2. We also trained variants with different network architectures, loss functions, and noise strategies. We reduce the channels from 512 to 256 of the last two layers denoted as *reduced channels*. We also changed the loss function from MSE to L1 and used the default standard Gaussian noise.

Table 8: Ablation Studies on Hyperparameters and Network Architectures.

| Methods | RMSE ↓ | REL ↓ | MAE ↓ | $\delta_{1.05}$ ↑ | $\delta_{1.10}$ ↑ | $\delta_{1.25}$ ↑ |
|---|---|---|---|---|---|---|
| Baseline | **0.0040** | 0.0014 | **0.0010** | **99.71** | **99.90** | **99.99** |
| $D^3$RoMa (reduced channels) | 0.0048 | 0.0016 | 0.0011 | 99.60 | 99.85 | 99.98 |
| $D^3$RoMa (L1 loss) | 0.0047 | **0.0008** | 0.0012 | 99.60 | 99.83 | 99.98 |
| $D^3$RoMa (randn noise) | 0.0048 | 0.0017 | 0.0012 | 99.64 | 99.87 | 99.98 |

## F.3   Ablations on Different Samplers and Inference Time

The main factors of run time are the input image resolution and the number of denoising steps. To better understand the computation-accuracy tradeoff, we list the runtime of our method and other SoTAs in Table 9. We also evaluate the effects of different samplers in the real experiments where we used pretrained StableDiffusion [67]. We report the inference time of our method in Table 10. The network has about 865M parameters in total for this variant. We fixed the number of time steps during training to 1000 and used the same standard cosine scheduler [66], and all the samplers take 10 denoising steps during inference. We perform reverse sampling using different schedulers implemented by Diffusers [65]. All the samplers achieve similar qualitative results except *Euler Ancestral*. The results are shown in Figure 8. Empirically, we select DDPM with 10 denoising steps and a resolution of 640×360 in our real experiments.

Table 9: **Runtime and memory consumption comparisons.** Our method has 113M parameters and takes 10 denoising steps during inference.

| Methods | NLSPN | LIDF | SwinDR | ASGrasp | IGEV-Stereo | Croco-Stereo | $D^3$RoMa |
|---|---|---|---|---|---|---|---|
| Runtime | 15ms | 37ms | 10ms | 40ms | 162ms | 138ms | 670ms |
| Peak Memory Usage | 4.79G | 1.16G | 1.07G | 3.14G | 2.84G | 2.78G | 2.81G |

Table 10: Runtime and memory consumption of our method during reverse sampling for single input. All times are reported on NVIDIA A100.

| Disparity Resolutions | 1280×720 | 640×360 | 480×270 | 320×180 | 224×126 |
|---|---|---|---|---|---|
| 5 Denoising steps | 5.53 | 2.56 | 2.31 | 1.95 | 1.96 |
| 10 Denoising Steps | 8.82 | 3.19 | 2.91 | 2.25 | 2.17 |
| 50 Denoising Steps | 34.56 | 8.45 | 5.79 | 4.28 | 3.86 |
| Peak Memory Usage | 18.62G | 7.87G | 7.57G | 6.94G | 6.89G |

## G   Alternative Guidance with Raw Disparity

This section will study alternative guidance to the diffusion model during the reverse sampling processes. In the stereo vision case, the gradient of the photometric loss is obtained by checking the consistency of the left and right images. Mathematically, the gradient should also have the same direction with $x_0 - x_t$ where $x_0$ is the ground truth disparity. In test time, $x_0$ is unknown but can be approximated by an external less noisy measurement source such as a Lidar. The external depth measurement can be converted to the disparity space $\tilde{x}_0$ and is multiplied with a mask if it is sparse:

$$\nabla_{x_t} \log p(x_t|y) = \nabla_{x_t} \log p(x_t|I_c) + \nabla_{x_t} \log p(I_r|I_l, x_t)$$
$$\approx \nabla_{x_t} \log p(x_t|I_c) + \alpha\omega(u,v)\text{sign}(\tilde{x}_0 - x_t) \tag{18}$$

We here experiment with raw depth guidance. The guidance $\tilde{x}_0$ is approximated by camera raw sensor depth. Therefore the $\text{sign}(\tilde{x}_0 - x_t)$ is the approximate gradient. We set mask $\omega(u,v) = (\tilde{x}_0 > 0)$ and $\alpha$ is a constant controls the guidance strength. We qualitatively study the guidance of the approximate gradient in Figure 9. The benefits of guidance by the approximate gradient are limited when raw depth is highly noisy.

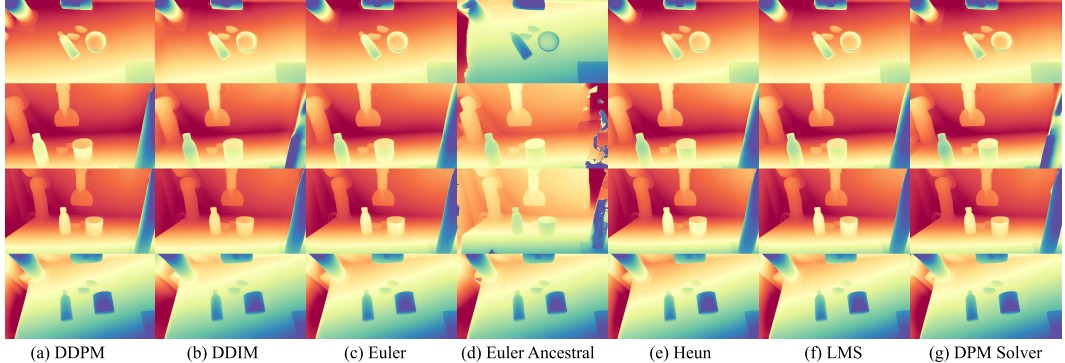

| (a) DDPM | (b) DDIM | (c) Euler | (d) Euler Ancestral | (e) Heun | (f) LMS | (g) DPM Solver |

Figure 8: Comparisons of different samplers used during reverse sampling. We use DDPM [30] sampler with 10 steps in the experiments.

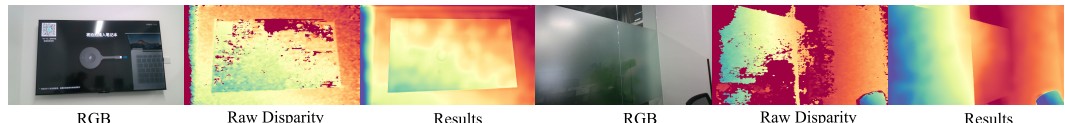

| RGB | Raw Disparity | Results | RGB | Raw Disparity | Results |

Figure 9: Guidance with raw disparity.

# H   More Experimental Results

## H.1   Detailed Comparisons on ClearPose Dataset

We provide the results of our method on ClearPose compared with two other baselines as shown in Table 11. There are a total of 6 different scenes, including different backgrounds, heavy occlusions, objects in translucent and opaque covers, on non-planar surfaces, and even filled with liquid.

Table 11: **Results on Depth completion benchmark ClearPose.** Our method $D^3$RoMa consistently outperforms both ImplicitDepth and TransCG on 6 different test scenarios.

| Testset | Metric | RMSE↓ | REL↓ | MAE↓ | $\delta_{1.05}$ ↑ | $\delta_{1.10}$ ↑ | $\delta_{1.25}$ ↑ |
|---|---|---|---|---|---|---|---|
| | ImplicitDepth | 0.07 | 0.05 | 0.04 | 67.00 | 87.03 | 97.50 |
| New Background | TransCG | **0.03** | 0.03 | 0.02 | 86.50 | 97.02 | 99.74 |
| | $D^3$RoMa(Cond. On RGB+Raw) | 0.05 | **0.01** | **0.01** | **96.71** | **98.84** | **99.74** |
| | ImplicitDepth | 0.11 | 0.09 | 0.08 | 41.43 | 66.52 | 91.96 |
| Heavy Occlusion | TransCG | **0.06** | 0.04 | **0.04** | 72.03 | 90.61 | 98.73 |
| | $D^3$RoMa(Cond. On RGB+Raw) | 0.10 | **0.03** | **0.04** | **83.97** | **93.69** | **98.79** |
| | ImplicitDepth | 0.16 | 0.16 | 0.13 | 22.85 | 41.17 | 73.11 |
| Translucent Cover | TransCG | 0.16 | 0.15 | 0.14 | 23.44 | 39.75 | 67.56 |
| | $D^3$RoMa(Cond. On RGB+Raw) | **0.13** | **0.06** | **0.07** | **63.07** | **82.78** | **95.80** |
| | ImplicitDepth | 0.14 | 0.13 | 0.10 | 34.41 | 55.59 | 83.23 |
| Opaque Distractor | TransCG | 0.08 | 0.06 | 0.06 | 52.43 | 75.52 | 97.53 |
| | $D^3$RoMa(Cond. On RGB+Raw) | **0.11** | **0.03** | **0.05** | **82.46** | **91.46** | **97.97** |
| | ImplicitDepth | 0.14 | 0.13 | 0.11 | 32.84 | 53.44 | 84.84 |
| Filled Liquid | TransCG | **0.04** | 0.04 | **0.03** | 77.65 | 93.81 | **99.50** |
| | $D^3$RoMa(Cond. On RGB+Raw) | 0.09 | **0.03** | **0.04** | **87.58** | **94.85** | 99.15 |
| | ImplicitDepth | 0.18 | 0.16 | 0.15 | 20.34 | 38.57 | 74.02 |
| Non Planar | TransCG | 0.09 | 0.07 | 0.07 | 55.31 | 76.47 | 94.88 |
| | $D^3$RoMa(Cond. On RGB+Raw) | **0.08** | **0.03** | **0.04** | **84.67** | **92.85** | **98.21** |

## H.2   Results on HISS Test Split

In this section, we train other SOTA stereo methods from scratch and compare with our method on the HISS dataset. We further rendered 300 images in 5 new scenes different from our training dataset as the test set. The results are given in Table 12. We also show more real depth estimation results in Figure 12 and more comparisons in Figure 13, which we consider also attributed to the joint training on our dataset.

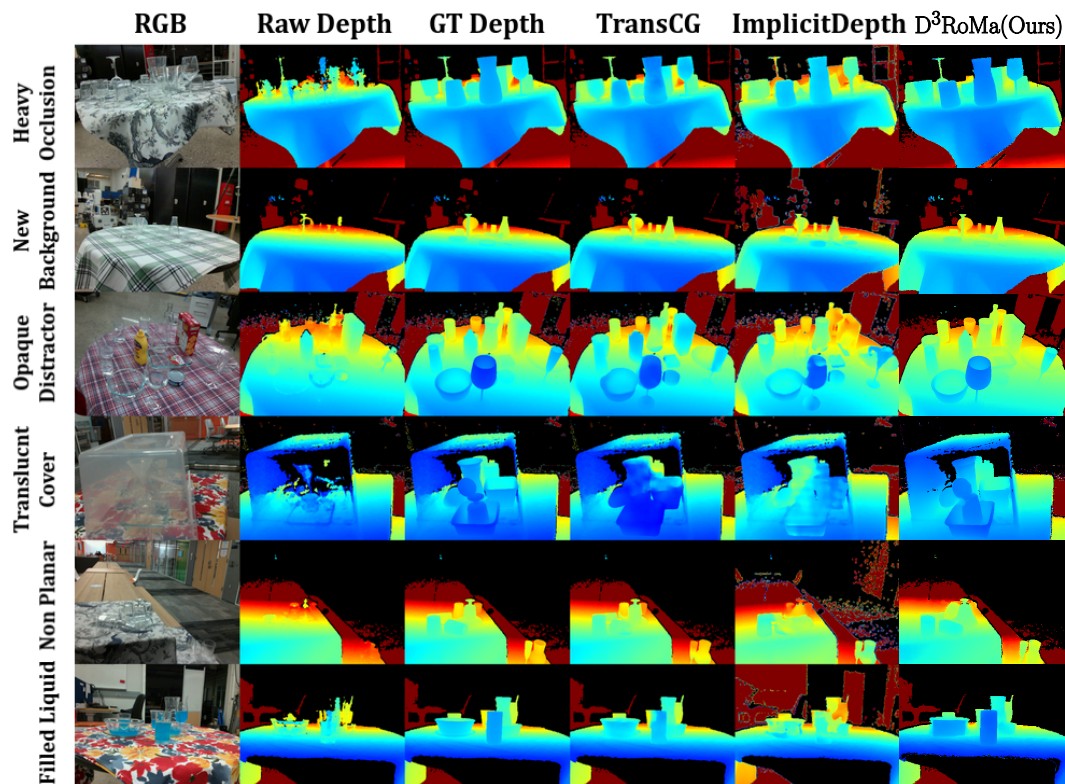

Figure 10: **Qualitative depth completion results on ClearPose.** From left to right, there are RGB image, raw depth, ground truth depth rendered using object CAD models, completed depth by TransCG, ImplicitDepth, and our method D³RoMa.

Table 12: Quantatives evaluations on HISS dataset.

| Methods | EPE | RMSE↓ | REL↓ | MAE↓ | $\delta_{1.05}$ ↑ | $\delta_{1.10}$ ↑ | $\delta_{1.25}$ ↑ |
|---|---|---|---|---|---|---|---|
| Raft-Stereo | 0.0721 | 0.0521 | 0.0092 | 0.0164 | 95.26 | 98.89 | 99.10 |
| D³RoMa | 0.0579 | 0.0378 | 0.0067 | 0.0084 | 97.86 | 99.22 | 99.76 |

## H.3  Deonising Process

One of the motivations for using the diffusion model to predict depth is the multi-step reverse sampling process. It resembles the iterative solver which has been proven successful in RAFT [6] and its successors. In figure 11 we show an example of the denoising process trained on our HISS dataset. The total denoising steps is set to 128 and we visualize every 32 timesteps.

## I  HISS Dataset

**HISS.**   We further evaluate the effectiveness of our dataset for transparent and specular object depth estimation. We compare our method with the previous state-of-the-art methods. As shown in Figure 13, compared with RAFT-Stereo [7], which is trained on large-scale datasets for stereomatching, our method can predict better depth, especially on transparent bottles. To ensure fair comparisons, we further fine-tune RAFT-Stereo on HISS for 400,000 epochs. Compared to the original model, the fine-tuned RAFT-Stereo can recover the missing depth of transparent objects better but the object shapes are still inaccurate. We also compare our method with ASGrasp [14] which is specially designed to detect and grasp transparent objects based on depth estimation. It has a similar performance to the fine-tuned RAFT-Stereo but has blurred object boundaries. Our methods can provide the best depth for all STD objects, with significantly clearer object boundaries

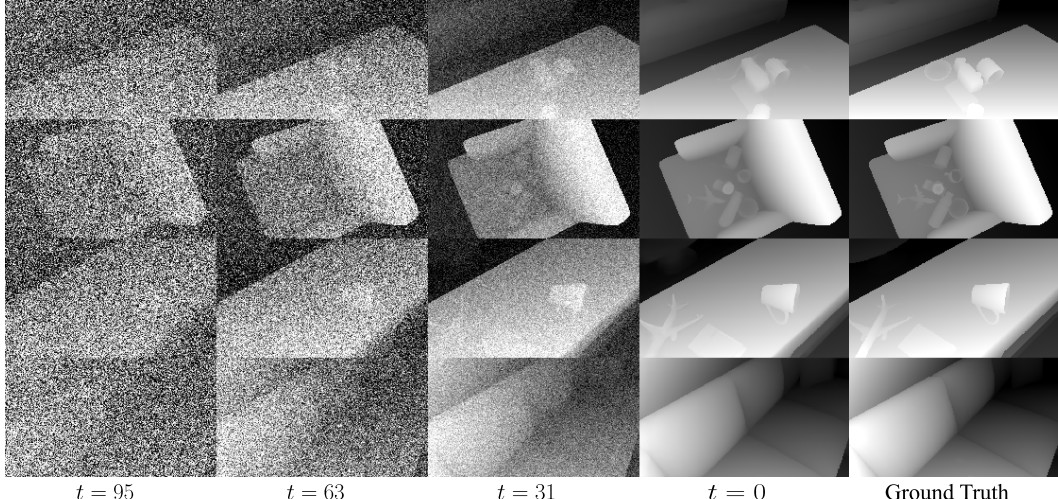

| $t = 95$ | $t = 63$ | $t = 31$ | $t = 0$ | Ground Truth |

Figure 11: Visualization of the denoising process on the HISS dataset. The 4 left columns show the denoising steps every 2 time steps. The 2 right columns show the final output and ground truth disparity map respectively.

and accurate shapes. More quantitative and qualitative results for the effectiveness of our dataset are provided in Appendix H.

One aspect that characterizes our dataset is *scene-level* and *photo-realistic* rendering of the *specular*, *transparent*, and *diffuse* objects. We rendered over 350 objects in 168 different HSSD [10] scenes. The objects randomly fall onto the furniture, ground, and tables to simulate real-world object placements. The infrared (IR) images are rendered properly with seeing-through or specular lighting effects on Non-Lambertian surfaces. We provide some data samples in Figure 14.

## J   Part Manipulation

### J.1   Interaction Policy

Following [35, 36], we first do part segmentation and pose estimation using the perception method. Based on the predictions of the part poses, we move the robot arm toward the target part and turn the gripper in the direction suitable for grabbing. Finally, we move the gripper along the proposed trajectories toward the target position, following our GAPart pose definition.

### J.2   Experiment Setup

In the experiments, we use the Franka Emika Panda robot arm with CuRobo[63] motion planning and the end-effector trajectory just like GAPartNet[35]. For manipulation tasks in the real world, a partial point cloud of the target object instance is acquired from our method. With the proposed network and manipulation heuristics in [35], the pose trajectory of the end-effector can be predicted. Then we use cuRobo[63] to solve the pose of Franka to follow our end-effector trajectory.

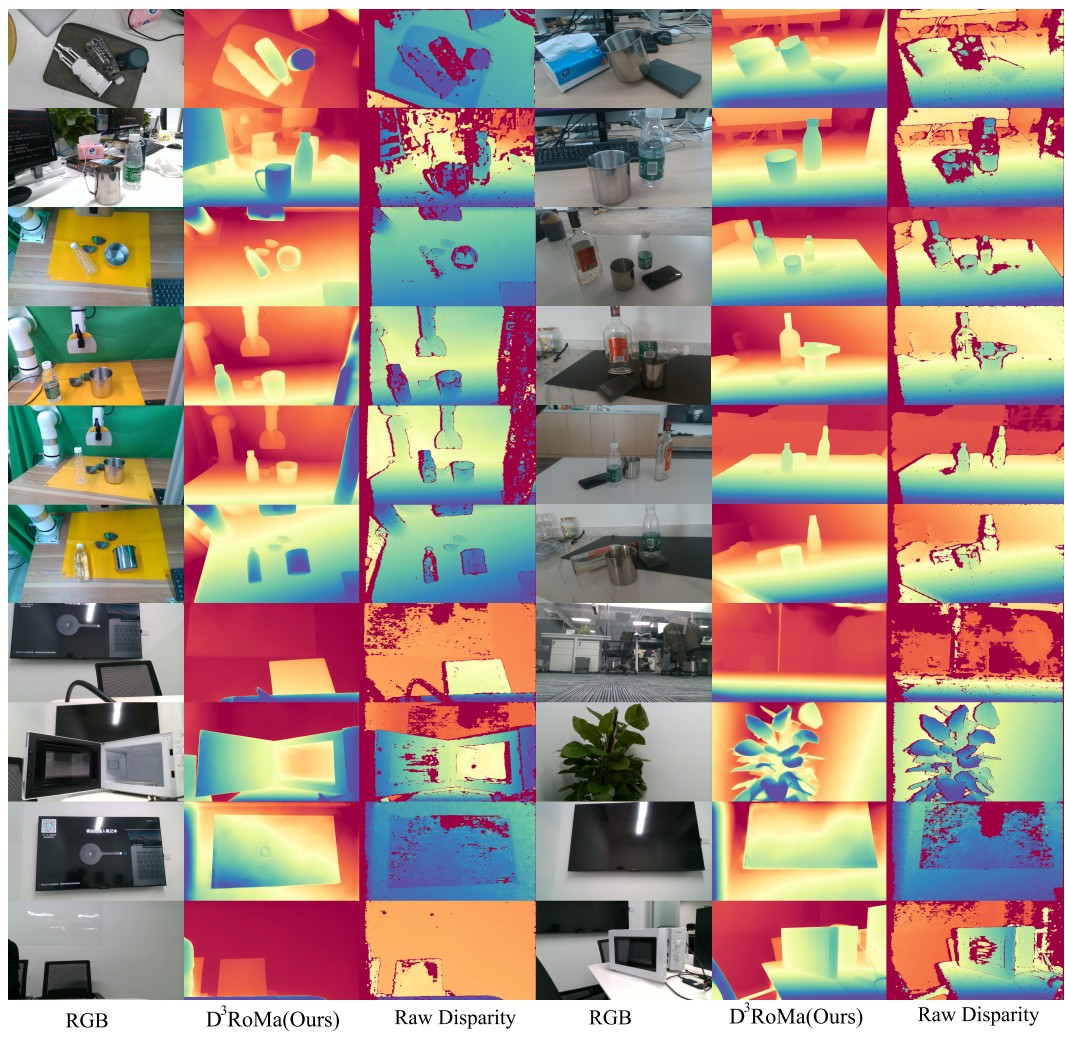

RGB     D³RoMa(Ours)     Raw Disparity     RGB     D³RoMa(Ours)     Raw Disparity

Figure 12: **More in-the-wild examples.** Each example consists of an RGB image, a raw disparity map, and our prediction.

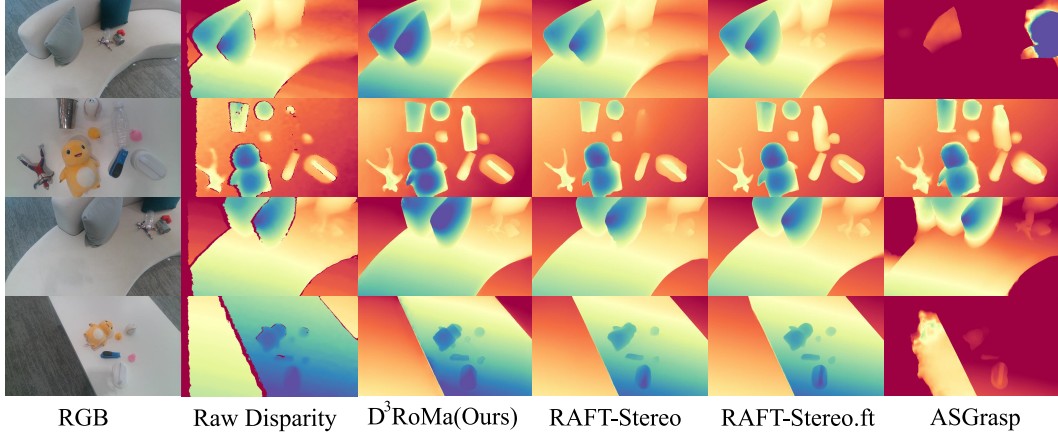

RGB     Raw Disparity     D³RoMa(Ours)     RAFT-Stereo     RAFT-Stereo.ft     ASGrasp

Figure 13: **Qualitative comparisons with other state-of-the-arts.**

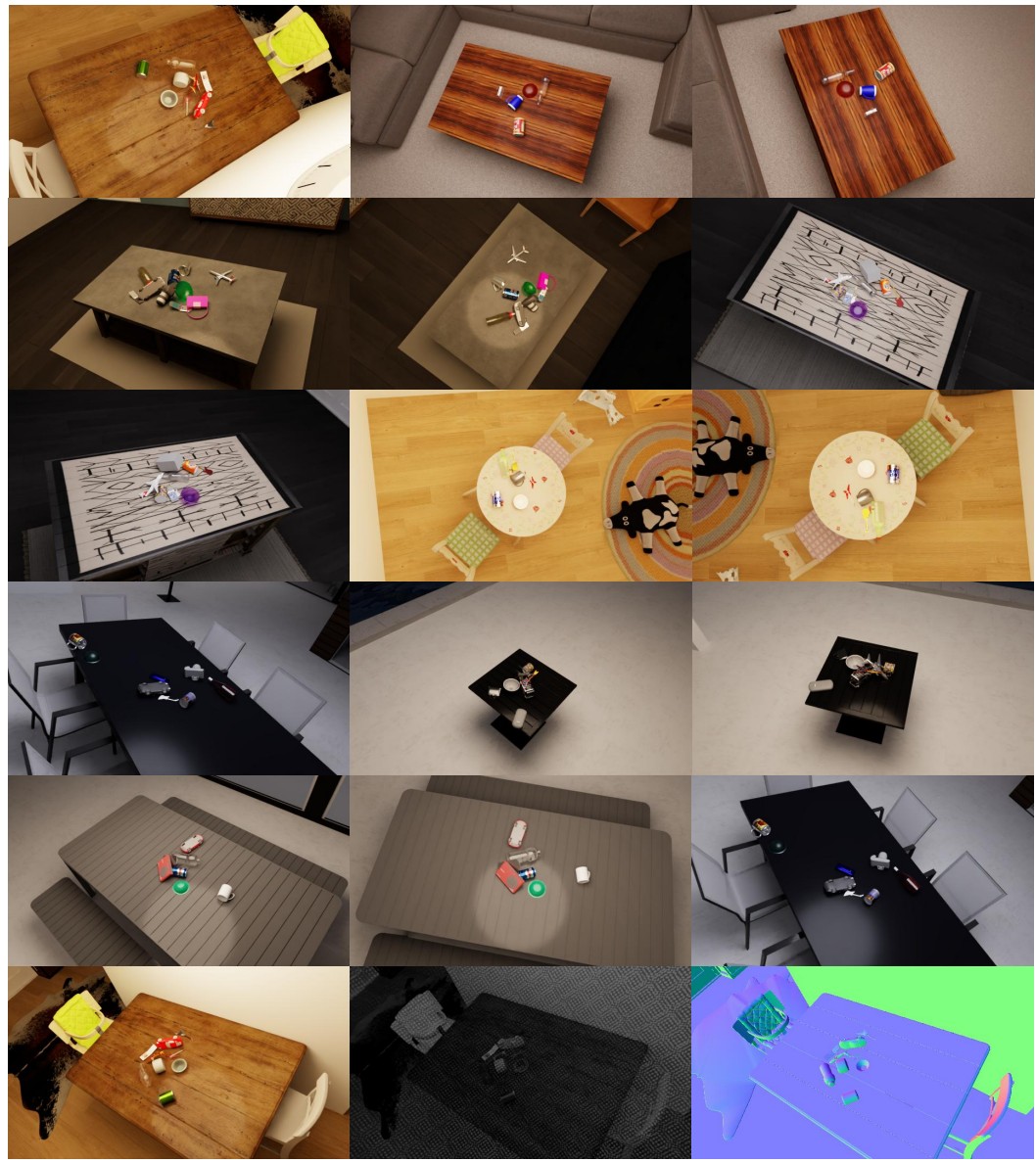

Figure 14: RGB Data samples from Our dataset HISS except the bottom row which shows a group of rendering of RGB image, (left) IR image, and normal.

