# OpenReview forum: "D$^3$RoMa: Disparity Diffusion-based Depth Sensing for Material-Agnostic Robotic Manipulation"
_robot-learning.org/CoRL/2024/Conference — CoRL 2024_

### Official Review · Reviewer_Jtaw · 2024-07-20
**review for D3RoMa**

**Originality:** 3
**Technical Quality:** 3
**Clarity Of Presentation:** 4
**Potential Impact:** 3
**Recommendation:** 3
**Confidence:** 4

**Review:**

Depth estimation is a crucial skill for robot control with vision feedback. However, existing depth sensor often reports incorrect raw depth, which limits robot performance. This paper proposes a new method for depth estimation using stereo image pair and raw disparity map either from sensor or traditional model peridtion. It uses a diffusion based approach to iteratively improve the disparity map conditioned on the stereo image pair. The performance of the proposed method is been validated on both real-world and simulated datasets. And it also perform well on real robot for vision feedback based manipulation task.  On top of that, experiment results show that the model is effective for challenging cases where objects are transparent or highly reflective. And a new synthetic dataset, HISS is created for training and testing the proposed method.

However, there are still several questions about the method and dataset introduced in this paper. Firstly one of the novelty claimed in this paper is using diffusion model for disparity prediction. And the author considers this method to be an image-to-image translation instead of disparity prediction and claims this to be another novelty point. However, using end to end model to predict whole disparity map is not a new idea and can be found in many publications. claiming that to be a novel idea is questionable. Furthermore, using diffusion model for disparity/depth prediction is also not new, papers like DiffusionDepth and DDP already shows diffusion model can be used for this task.

Another question is related to the novelty of HISS dataset, compared to existing transparent object datasets, both real and rendered, it is lacking in terms of number of objects and scenes. What is the necessity to create a new dataset when existing dataset like ClearPose and SynTODD are already there.

**Quality Of The Limitations Section:**

3

**Questions For Rebuttal:**

1. justify why using diffusion for disparity prediction is novel when existing works already did depth prediction using diffusion.
2. what is the impact of the original disparity map on the final prediction, especially when raw map is wrong or incomplete for transparent and specular objects. what is the difference between disparity from sensor and from other model prediction?
3. For rendering synthetic dataset, what is the method used to mimic the raw disparity? how do you make sure the simulated raw disparity is a good estimation of real-world, especially for transparent and specular objects.
4. justify why not using existing transparent objects datasets instead of creating a new one, especially when existing datasets are better.
5. Provide your model performance on existing transparent object datasets like ClearPose and SynTODD.
6. For the claimed sim2real performance, report model eval result on ClearPose

**Robotics Focus:**

4

**Summary Of Paper:**

Depth estimation is a crucial skill for robot control with vision feedback. However, existing depth sensor often reports incorrect raw depth, which limits robot performance. This paper proposes a new method for depth estimation using stereo image pair and raw disparity map either from sensor or traditional model peridtion. It uses a diffusion based approach to iteratively improve the disparity map conditioned on the stereo image pair. The performance of the proposed method is been validated on both real-world and simulated datasets. And it also perform well on real robot for vision feedback based manipulation task.  On top of that, experiment results show that the model is effective for challenging cases where objects are transparent or highly reflective. And a new synthetic dataset, HISS is created for training and testing the proposed method.

**Summary Of Recommendation:**

Depth estimation is a crucial skill for robot control with vision feedback. However, existing depth sensor often reports incorrect raw depth, which limits robot performance. This paper proposes a new method for depth estimation using stereo image pair and raw disparity map either from sensor or traditional model peridtion. It uses a diffusion based approach to iteratively improve the disparity map conditioned on the stereo image pair. The performance of the proposed method is been validated on both real-world and simulated datasets. And it also perform well on real robot for vision feedback based manipulation task.  On top of that, experiment results show that the model is effective for challenging cases where objects are transparent or highly reflective. And a new synthetic dataset, HISS is created for training and testing the proposed method.  However, there are still several questions about the method and dataset introduced in this paper. Firstly one of the novelty claimed in this paper is using diffusion model for disparity prediction. And the author considers this method to be an image-to-image translation instead of disparity prediction and claims this to be another novelty point. However, using end to end model to predict whole disparity map is not a new idea and can be found in many publications. claiming that to be a novel idea is questionable. Furthermore, using diffusion model for disparity/depth prediction is also not new, papers like DiffusionDepth and DDP already shows diffusion model can be used for this task.   Another question is related to the novelty of HISS dataset, compared to existing transparent object datasets, both real and rendered, it is lacking in terms of number of objects and scenes. What is the necessity to create a new dataset when existing dataset like ClearPose and SynTODD are already there.

---

### Official Review · Reviewer_jawu · 2024-07-22
**Impressive performance with diffusion and new dataset**

**Originality:** 3
**Technical Quality:** 4
**Clarity Of Presentation:** 4
**Potential Impact:** 4
**Recommendation:** 4
**Confidence:** 4

**Review:**

This work addresses the important but understudied problem of depth perception of transparent and reflective objects using a stereo camera. This is a task relevant to robotics as depth cameras are widely available and tend to provide more robust scale than monocular approaches. The contribution consists of a new dataset, which builds on DREDS to get simulated RGBD even for transparent and reflective objects. The authors improve the dataset by adding more generic scenes, as well as the transparent and reflective assets from DREDS.

On the method side, the authors introduce a geometry-aware loss to guide the diffusion and achieve better results - although I have not seen this ablated. The simulation results show D3RoMa is competitive against Raft-Stereo when exposed to the same data, although it would have been better to evaluate more SoA stereo methods in the same way. A limiting factor in evaluation is the lack of other stereo datasets for transparent objects.

The robot experiments are great, and show the improvement in performance in the real world beyond the training set. Perhaps you could add some of the sensory readings of these experiments of the appendix? This would help explain the challenges in each of the subsets of the tasks. For example, the performance in figure 8 in the appendix is convincing.

Overall the authors have contributed a method for stereo mvs of transparent and reflective objects, a usually understudied subject. They have done so with significant improvements over SoA, a new method and dataset.

**Quality Of The Limitations Section:**

3

**Questions For Rebuttal:**

See comments above, consider adding ablations and more information/data on the real-world experiments. Also consider adding mode limitations and failure cases in the limitations section.

**Robotics Focus:**

4

**Summary Of Paper:**

This paper addresses the problem of depth perception of transparent objects with a stereo camera. This is an understudied topic, usually monocular or depth completion is the focus. The contributions are: (1) a diffusion-based stereo depth estimation frame-work specifically designed for transparent and reflective objects. On the method side, a geometry-guided diffusion, a new synthetic dataset, evaluation on robot tasks.

**Summary Of Recommendation:**

Sufficient novelty and improvement over SoA.

---

### Official Review · Reviewer_SQrs · 2024-07-31
**Review for paper titled D3RoMa: Disparity Diffusion-based Depth Sensing for Material-Agnostic Robotic Manipulation**

**Originality:** 4
**Technical Quality:** 4
**Clarity Of Presentation:** 4
**Potential Impact:** 3
**Recommendation:** 4
**Confidence:** 4

**Review:**

**Strengths:**
1. This paper handles an important problem of estimating depth for translucent objects with an interesting approach.
2. The evaluation of the proposed method is done on various tasks and configurations and provides a clear improvement over the state of the art.
3. Good ablation studies are performed to show the effectiveness of various configurations and the proposed design choices.

**Concerns/weaknesses:**
1. The accurate depth results for translucent objects currently seem like a happy result of the proposed black-box diffusion model. I request the authors to comment on how the diffusion models can solve this problem compared to the cost-volume-based approaches.

2. I can only see run time evaluations of the proposed method but, it would be nice to have the comparison with the other methods from Table 1 to understand the trade-offs between accuracy, speed, and memory.
3. Please explain why adding disparity resulted in better convergence and results. Also, please comment on how this improved the depth estimation of translucent objects.
4. Please comment on what happens when the pre-processed disparity is bad. Did you perform any evaluation on progressively destroying the disparity by adding noise or masking?
5. I see the quantitative results for RAFT-Stereo in the sup-material. Please try to add them to the main paper for better understanding and readability.

Please address my concerns in the rebuttal response.

**Quality Of The Limitations Section:**

2

**Questions For Rebuttal:**

Please see the concerns/weaknesses above.

**Robotics Focus:**

4

**Summary Of Paper:**

This paper proposes a denoising diffusion probabilistic model method to estimate accurate depth, even for translucent objects from stereo images for indoor scenes by casting the image restoration and depth estimation problem into an image-to-image translation problem. They collected a synthetic dataset with specular objects to train and better evaluate the proposed method. Various experiments are performed in both simulation and the real world to prove the effectiveness of the proposed method.

**Summary Of Recommendation:**

Overall this is a well written paper with good evaluation and ablation studies. However, I have some concerns and questions. I would like to see the response from the authors during the rebuttal process.

---

### Author Rebuttal · Authors · 2024-08-10

Please note the attached zip file contains:

1. A revised paper that addresses the concerns of each reviewer.
2. Some point clouds which are obtained by back-projecting depths from the raw sensor and our prediction. The *.ply files can be opened by MeshLab.
3. All the revised figures and tables.

---

### Decision · Program_Chairs · 2024-09-04

**Decision:**

Accept

**Comment:**

Summary of Strengths:
* Important problem of estimating depth for translucent objects with an interesting approach
* A new dataset
* Experiment with a real robot
*  Clear improvement over SOTA

Summary of Weaknesses:
* Comparison with other SOTA - addressed  and to be added to the paper
* Needs to clarify novelty and parts of the method -  addressed and to be added to the paper
* Missing explanation about why it works on translucent objects -  addressed and to be added to the paper